# Management of Kidney Failure in Patients with Diabetes Mellitus: What Are the Best Options?

**DOI:** 10.3390/jcm10132943

**Published:** 2021-06-30

**Authors:** Juan M. Buades, Lourdes Craver, Maria Dolores Del Pino, Mario Prieto-Velasco, Juan C. Ruiz, Mercedes Salgueira, Patricia de Sequera, Nicanor Vega

**Affiliations:** 1Department of Nephrology, Hospital Universitario Son Llàtzer, Balearic Islands, 07198 Palma de Mallorca, Spain; 2Health Research Institute of the Balearic Islands (IdISBa), 07120 Palma de Mallorca, Spain; 3Department of Nephrology, Hospital Universitario Arnau de Vilanova, 25198 Lleida, Spain; lscraver.lleida.ics@gencat.cat; 4Department of Nephrology, Complejo Hospitalario Torrecárdenas de Almería, 04009 Almería, Spain; mdpinoypino@gmail.com; 5Department of Nephrology, Complejo Asistencial Universitario de Leon, 24001 León, Spain; mprietov@gmail.com; 6Department of Nephrology, Valdecilla Hospital, University of Cantabria, 39008 Santander, Spain; juancarlos.ruiz@scsalud.es; 7Valdecilla Biomedical Research Institute (IDIVAL), Cardenal Herrera Oria S/N, 39011 Santander, Spain; 8Department of Nephrology, Hospital Universitario Virgen Macarena, 41009 Seville, Spain; salgueiramer@gmail.com; 9Biomedical Engineering Group, Medicine Department, University of Seville, 41092 Seville, Spain; 10Center for Biomedical Research Network in Bioengineering Biomaterials and Nanomedicina (CIBER-BBN), 28029 Madrid, Spain; 11Department of Nephrology, Hospital Universitario Infanta Leonor, 28031 Madrid, Spain; psequerao@senefro.org; 12Medicine Department, Universidad Complutense de Madrid, 28031 Madrid, Spain; 13Department of Nephrology, Hospital Universitario de Gran Canaria Dr. Negrín, 35010 Las Palmas de Gran Canaria, Spain; nvegdia@gobiernodecanarias.org

**Keywords:** chronic kidney disease, diabetic kidney disease, hemodialysis, peritoneal dialysis, home hemodialysis, kidney transplant, kidney failure, kidney replacement therapy, comprehensive conservative care

## Abstract

Diabetic kidney disease (DKD) is the most frequent cause of kidney failure (KF). There are large variations in the incidence rates of kidney replacement therapy (KRT). Late referral to nephrology services has been associated with an increased risk of adverse outcomes. In many countries, when patients reach severely reduced glomerular filtration rate (GFR), they are managed by multidisciplinary teams led by nephrologists. In these clinics, efforts will continue to halt chronic kidney disease (CKD) progression and to prevent cardiovascular mortality and morbidity. In patients with diabetes and severely reduced GFR and KF, treating hyperglycemia is a challenge, since some drugs are contraindicated and most of them require dose adjustments. Even more, a decision-making process will help in deciding whether the patient would prefer comprehensive conservative care or KRT. On many occasions, this decision will be conditioned by diabetes mellitus itself. Effective education should cover the necessary information for the patient and family to answer these questions: 1. Should I go for KRT or not? 2. If the answer is KRT, dialysis and/or transplantation? 3. Dialysis at home or in center? 4. If dialysis at home, peritoneal dialysis or home hemodialysis? 5. If transplantation is desired, discuss the options of whether the donation would be from a living or deceased donor. This review addresses the determinant factors with an impact on DKD, aiming to shed light on the specific needs that arise in the management and recommendations on how to achieve a comprehensive approach to the diabetic patient with chronic kidney disease.

## 1. Introduction

Chronic kidney disease (CKD), defined as the presence of markers of kidney damage or a glomerular filtration rate (GFR) <60 mL/min/1.73 m^2^, for three months or more, irrespective of cause, is classified according to the categories of GFR and albuminuria [1]. Diabetic kidney disease (DKD) is described as the persistent presence of kidney injury (either by impaired glomerular filtration rate, albuminuria or histological alterations) in subjects with diabetes mellitus (DM), in the absence of signs of other forms of kidney disease. Diabetic kidney disease is a heterogeneous disease, which includes multiple and complex overlapping etiologic pathways. Its hallmarks include alterations in glomerular hemodynamics, extracellular matrix synthesis/degradation balance, inflammation, interstitial fibrosis, oxidative stress, and tubular atrophy.

Patients with DKD have an eight-fold increased risk of cardiovascular and all-cause mortality compared to those without DM and CKD [2,3,4]. Furthermore, DKD is the most frequent cause of kidney failure with replacement therapy (KFRT) [5], which denotes an important morbidity of this disease. Following the recent suggestions of the KDIGO Consensus Conference [6], when we refer to kidney failure (KF) we defined it as a GFR <15 mL/min per 1.73 m^2^ or treatment by dialysis. Kidney replacement therapy (KRT) may consist of either the realization of a renal transplant or the initiation of dialysis therapy.

For the evaluation of those patients who are candidates to receive a renal transplant, there are guidelines to guide the decision making process regarding the convenience of transplantation and the appropriate time to perform it [7]. On the other hand, deciding the appropriate time to initiate dialysis treatment is not a simple task as there are a number of factors that must be taken into account. It is widely accepted that the GFR level alone should not be used as a reason to start KRT, and that signs and symptoms associated with KF should be considered [8].

Based on the above, it is important to educate the healthcare community about the benefits of a referral protocol for patients with progressive kidney function decline, since late referral to nephrology services has been associated with an increased risk of adverse outcomes [9]. For instance, in many countries, when patients present with severely reduced GFR (CKD G4), they are evaluated by multidisciplinary teams led by nephrologists. The importance of this approach is to try to predict which patients have a greater risk of mortality and which are more likely to progress to KF. The multidisciplinary strategy seeks to halt the progression of CKD and to intensify the prevention of mortality and cardiovascular morbidity. In addition to these, the control of factors associated with kidney disease which play a role in its course and prognosis (i.e., anemia, bone and mineral disorders, hydro-electrolytic alterations) is becoming more relevant. Glycemic control in diabetic patients with CKD presents an added difficulty due to the need for dose adjustment of several hypoglycemic drugs, including contraindications in some cases.

In this review we will address the factors that influence CKD in the patient with DM, including those that predict the occurrence of cardiovascular events and progression to kidney failure in DKD. The aim is to shed light on the specific needs that arise in the management of DKD and recommendations on how to achieve a comprehensive approach, either conservatively or through KRT. For this, the role of the nursing team as well as the collaboration between nephrologists, endocrinologists, vascular surgeons, general surgeons, interventional radiologists, nutritionists, psychologists and physical exercise specialists, among others, will be of great importance.

## 2. Delaying the Progression of Diabetic Kidney Disease in Patients with CKD G4

Due to its complex pathogenesis, DKD presents a wide clinical variability. The incidence rate of KF in type 2 DM (T2DM) is 0.29% at 10 years and 0.74% at 20 years from the diagnosis of diabetes [10], while for type 1 diabetes mellitus (T1DM) it is 2.5 per 1000 person-years [11]. However, 40%–53% of diabetics have mild renal disease [12,13]. Additionally, the rate of estimated glomerular filtration rate (eGFR) decline is more rapid (loss > 3 mL/min/1.73 m^2^ per year) in DM than in healthy individuals, particularly in patients with a long duration of diabetes (more than 10 years), severely increased albuminuria, or low baseline eGFR [14]. Still, variability is high. A previous study analyzing the evolution of kidney function in T2DM showed that 28% of patients presented no decline, 56% of patients showed a moderate decline (−4 mL/min/1.73 m^2^ ≤ annual decline < 0 mL/min/1.73 m^2^) and 15% of them presented severe decline (annual decline < −4 mL/min/1.73 m^2^) [12].

Unfortunately, the traditionally employed clinical biomarkers, although very useful in the diagnosis and long-term follow-up of the disease, are usually not sensitive enough to detect early kidney injury. To illustrate, the absence of albuminuria does not exclude the presence of diabetic kidney injury or subclinical renal damage [15]. On the other hand, a benefit in the treatment of albuminuria cannot be excluded either, despite not achieving the desired goal [16], since in CKD patients: (a) the risk of kidney events increases in association with moderately increased albuminuria (A2), and the increment is even further with severely increased albuminuria (A3); (b) the risk of cardiovascular events is elevated in those with moderately and severely increased albuminuria; and (c) all-cause mortality is increased in subjects with a GFR lower than 30 mL/min/1.73 m^2^ independently of albuminuria [17]. Moreover, DKD patients are more susceptible to acute kidney injury (AKI), which might contribute to interstitial fibrosis. In a cohort of 4082 patients with diabetes, single or repetitive episodes of AKI significantly increased the risk of developing advanced CKD [18].

Numerous risk factors (Table 1) have been identified for the development and progression of DKD. Hence, interindividual variability is the result of the interaction between sociodemographic and clinical risk factors, as well as adequate glycemic and blood pressure control.

### 2.1. Duration, Socioeconomic Factors and Familial and Genetic Factors

As mentioned above, one of the most determinant factors in CKD progression is the time course of diabetes. Cumulative risk of KFRT was 0.29% at 10 years and 0.74% at 20 years from diagnosis of T2DM [10]. Studies have demonstrated that a low socioeconomic status is associated with increased prevalence of DM, hypertension, and CKD [19].

The risk of developing DKD has a polygenetic component [20,21,22]. Diabetic patients with a first-degree relative with DKD have a substantially greater risk of developing DKD compared with those who do not have an affected relative [23]. Isolating a definitive causal pathway has proved to be elusive because there is no simple Mendelian inheritance and the interplay of several genes is likely involved and may differ between populations [24,25,26].

### 2.2. Hyperglycemia

Hyperglycemia is an important risk factor for the progression of albuminuria in diabetic subjects; however, its influence is more modest in the progression to kidney failure than the effect exerted by hypertension, hypercholesterolemia and genetic factors [14]. Among people with DM, studies suggest that the effect of more intensive glycemic control was associated with a reduction in albuminuria but it did not reduce significantly clinical kidney end-points, including doubling of serum creatinine, KFRT, and death from renal disease [27,28]. A review of the DOPPS study by Ramirez et al. showed a U-shaped association between HbA1c and all-cause mortality in DM-hemodialysis patients, showing the lowest death rates in association with high HbA1c levels (7 to 7.9%) [29]. The European Best Practice Guidelines (EBPG) advise against tighter glycemic control if this leads to severe hypoglycemic episodes, while at the same time it recommends tightening glycemic control when HbA1C values are >8.5% (69 mmol/mol) [30]. A high-quality systematic review demonstrated a lack of benefit of tighter glycemic control as assessed by an HbA1C <7% (53 mmol/mol) or 7.5% (59 mmol/mol) [31], whereas it rather resulted in a risk of hypoglycemia episodes. In CKD with severely reduced GFR or KF the risk of hypoglycemia increases, among other reasons, due to an alteration in pharmacological metabolism. In order to avoid this, patients at high risk of hypoglycemia should perform their own regular monitoring of blood glucose levels through validated devices. It is important to keep in mind that below an eGFR of 30 mL/min/1.73 m^2^, the shortening of erythrocyte life biases the measurement of low HbA1c, especially in patients receiving erythropoietin-stimulating agents.

The goals and treatments for the management of diabetes in CKD are described in the new KDIGO guidelines [32]. Cardiovascular prevention and avoiding CKD progression are the main objectives (cardio-nephroprotection). Most studies have been conducted in the early stages of DM and have excluded individuals with severely reduced GFR. In the recent years, different clinical trials have been published involving diverse pharmacological interventions with encouraging results, which have broadened their inclusion criteria for lower GFRs. The DAPA-CKD clinical trial [33] has demonstrated robust data on the role of sodium-glucose cotransporter-2 (SGLT2) inhibitors in slowing CKD progression in patients with chronic kidney disease (GFR 25–75 mL/min/1.73 m^2^), regardless of the presence or absence of diabetes, when comparing dapaglifozin with placebo (hazard ratio (HR), 0.61; 95% confidence interval (CI), 0.51 to 0.72; *p* < 0.001). Dapagliflozin has proven to be effective as a cardiovascular and renal protective treatment, showing a significant decrease in mortality, in the incidence of KFRT and in suffering a decline of at least 50% in the estimated GFR (HR, 0.56, 95% CI, 0.45 to 0.68, *p* < 0.001). Previously, the CREDENCE study [34] analyzed the effect of canagliflozin, another oral SGLT2 inhibitor, on renal outcomes in patients with T2DM and albuminuric CKD. The study demonstrated that canagliflozin was able to reduce by 30% the relative risk of CKD progression and cardiovascular death in diabetic patients compared with the placebo group (HR, 0.70; 95% CI, 0.59 to 0.82; *p* < 0.001) at a median follow-up of 2.62 years. These findings show that SGLT2 inhibitors can be an effective therapy option for renal and cardiovascular protection in patients with type 2 diabetes with chronic kidney disease. Additionally, a subgroup analysis of the CREDENCE trial [35] showed that canagliflozin reduced the risk of kidney disease progression to KF in participants with eGFR ≥ 30 mL/min/1.73 m^2^ (HR, 0.70; 95% CI, 0.54 to 0.91), whose effects were maintained in those subjects with an eGFR < 30 mL/min/1.73 m^2^ (HR, 0.67; 95% CI, 0.35 to 1.27; p interaction = 0.80). There was also no difference in the rate of kidney-related adverse events or AKI, associated with canagliflozin, between participants with eGFR < 30 and ≥30 mL/min/1.73 m^2^. The results support the use and continuation of SGLT2 inhibitors until initiation of dialysis or kidney transplantation (KT).

In summary, for adequate glycemic control in the diabetic patient with CKD, it is important to take into account the need for lifestyle therapy through physical exercise, nutritional intervention and weight loss. The KDIGO guidelines recommend as first-line antihyperglycemic therapy the use of metformin, as long as the GFR ≥ 30 mL/min/1.73 m^2^ [32]. Metformin exerts an effective HbA1c reduction with a safe profile due to the low risk of hypoglycemia [36,37,38]. In addition, it has been shown to reduce weight in obese patients, as well as the incidence of cardiovascular events. On the other hand, SGLT2 inhibitors are also considered first-line therapy for diabetic patients with GFR ≥ 30 mL/min/1.73 m^2^, either in combination with metformin or when metformin cannot be used [32]. Different trials [34,39,40,41] have proven the cardio- and nephroprotective role of these drugs, as well as their efficacy and safety in the presence of low GFR. The guidelines recommend not starting SGLT2 inhibitors when the GFR is below 30 mL/min/1.73 m^2^; however, in accordance with the approach followed in the CREDENCE trial [34], in those patients who are already taking it and have a drop in GFR below 30 mL/min/1.73 m^2^, the SGLT2 inhibitor can be continued until initiation of kidney replacement therapy. Furthermore, when diabetic patients with CKD require additional treatment for glycemic control, or cannot use metformin and/or SGLT2 inhibitors, the KDIGO guidelines recommend glucagon-like peptide-1 receptor agonists (GLP-1 RA) as the preferred option, due to the cardiovascular and renal benefit they have proven in the recent trials [42,43,44,45,46]. These recommendations are in line with the ACC [47], ADA [48] and ESC/EASD guidelines [3]. GLP-1 RA have shown a 36% to 15% reduction in the risk of CKD progression, especially at the expense of improved albuminuria control. These drugs can be used up to a GFR of 15 mL/min/1.73 m^2^ [49]. Options recommended by KDIGO guidelines for patients with GFR < 15 mL/min/1.73 m^2^ are DPP-4 inhibitors, insulin, and thiazolidinediones [32]. It is important to emphasize that the choice of the most appropriate treatment for each patient should be based on the patient’s preferences, comorbidities, eGFR (Table 2) and cost. Figure 1 summarizes the KDIGO Diabetes Management in CKD Guideline.

First-line antiglycemic medications (green) are metformin and SGLT2 inhibitor. As second-line therapy (purple) it is recommended to prioritize GLP-1 AR over other antiglycemic medications due to its cardiovascular and renal benefits. Other drugs that can be used are iDPP4, insulin, thiazolidinedione, sulfonylureas and alpha-glucosidase inhibitor. In patients with KF, iDPP4, insulin and thiazolidinedione are more suitable options. CKD, Chronic kidney disease; GFR, Glomerular filtration rate; SGLT2, Sodium-glucose cotransporter-2; GLP-1 RA, Glucagon-like peptide-1 receptor agonists; DPP-4, Dipeptidyl peptidase-4; KF, Kidney failure; KRT, Kidney replacement therapy; G1: GFR ≥90 mL/min per 1.73 m^2^; G2: GFR 60-89 mL/min per 1.73 m^2^, G3a: GFR 45–59 mL/min per 1.73 m^2^; G3b: GFR 30–44 mL/min per 1.73 m^2^; G4: GFR 15–29 mL/min per 1.73 m^2^; G5: GFR <15 mL/min per 1.73 m^2^ or treated by dialysis. (1) A. S. Levey et al., “Nomenclature for kidney function and disease: executive summary and glossary from a Kidney Disease: Improving Global Outcomes (KDIGO) consensus conference,” Journal of Nephrology, vol. 33, no. 4. pp. 639–648, 2020. (2) I. H. de Boer et al., “Executive summary of the 2020 KDIGO Diabetes Management in CKD Guideline: evidence-based advances in monitoring and treatment,” Kidney Int., vol. 98, no. 4, pp. 839–848, Octorber 2020.

### 2.3. Hypertension

Many authors recommend a blood pressure (BP) target of 140/90 mmHg for patients with DM, regardless of CKD [1]. Others suggest a target of 130/80 mmHg in the presence of moderately to severely increased albuminuria (A2 and A3) [50]. In patients aged 75 years or older, it is maintained at <150/90 mmHg regardless of GFR category (G1–G5) and the presence of DM, and at <140/90 mmHg if there are no adverse events such as orthostatic hypotension [51]. In a meta-analysis of 157 randomized controlled trials comparing BP-lowering agents in adults with type 2 DM and CKD, no blood pressure-lowering strategy was superior to placebo regarding survival. Treatment with angiotensin-converting enzyme inhibitors (ACEi) and angiotensin-II receptor blockers (ARB) showed the greatest efficacy for the prevention of KFRT; however, only the ARB was significantly better than placebo. The effects on BP did not differ between treatment regimens, demonstrating that pharmacological effects are independent of BP lowering. No regimen significantly increased hyperkalemia or AKI, although combined treatment with ACEi plus ARB led to borderline increases in estimated risks of these harms [52]. For patients who develop hyperkalemia, measures are available to control potassium levels, such as moderating potassium intake, diuretic initiation, use of sodium bicarbonate in those with metabolic acidosis, and co-administrating gastrointestinal cation exchangers.

Finally, and in reference to the RAAS axis blockade, in patients with an eGFR < 30 mL/min/1.73 m^2^, the European guidelines advise starting treatment with ACEi if there is a cardiac indication, but also advise to stop it if there are side effects. If renal function progresses to an eGFR < 15 mL/min/1.73 m^2^, due to the risk of a cardiovascular event or the initiation of dialysis, it is advisable to evaluate its interruption in an attempt to delay the start of dialysis [30].

Another drug of interest is finerenone (non-steroidal antagonist of the mineralocorticoid receptor). In the FIDELIO-DKD study, it shows a decrease in the incidence of cardiovascular events (HR, 0.86; 95% CI, 0.75 to 0.99; *p* = 0.034), as well as a decreased in kidney events (HR, 0.82; 95% CI, 0.73 to 0.93; *p* = 0.001) in patients with GFR 25–75 mL/min/1.73 m^2^ [53,54].

### 2.4. Lipids

People with diabetes and severely reduced GFR typically have significant hypertriglyceridemia, high LDL and low HDL cholesterol levels [55]. These abnormalities tend to be more pronounced when severe albuminuria is present, and they diminish with progression to KF and dialysis [14]. The SHARP study [56] (average eGFR 27 mL/min/1.73 m^2^) found that the association of statin plus ezetimibe significantly reduced major atherosclerotic events. The relative effect was similar when diabetes was present (23% of 9438 patients had diabetes). This study did not detect any effect of lipid-lowering therapy on the frequency of doubling of baseline serum creatinine concentration or progression to KFRT [57]. The role for dyslipidemia in the development and progression of DKD is unclear. Joint guidelines of the Association of British Clinical Diabetologists and the Renal Association recommend that hypolipidemic treatment with statins should be considered for all patients with CKD stages 3–5 and DKD (mainly because of cardiovascular risks). Fibrates should rarely be considered for treatment in this population [58]. No effect on CKD progression has been observed by the use of PCSK-9 inhibitors [59].

### 2.5. Diet

Long-term high protein intake accelerates structural and functional injury in models of DKD, whereas low protein provides kidney protection [14]. Nevertheless, there are no observational data in humans that unequivocally support this role of dietary protein. Yet it is accepted that a diet with uncontrolled intake of calories, protein, sodium, and phosphates exacerbates clinical metabolic alterations related to CKD G4–G5; hence, appropriate dietary-nutritional therapy may delay the need for KRT. It is known that G4–G5 CKD is characterized by a dysbiosis of the intestinal microbiota, which contributes to uremic intoxication and cardiovascular damage. Low-protein nutrition therapy associated with adequate fiber intake can counteract dysbiosis and reduce uremic toxins’ production [60]. Increasing evidence suggests that focusing on dietary intake patterns, rather than individual nutrient intake per se, offers an insightful approach to examine and identify the role of diet in CKD [61].

Adequate salt intake has been associated with reduced BP and improved control of albuminuria. In fact, sodium intake is the single most important factor in CKD progression and the control of hypertension. Dietary sodium is known to significantly modulate the nephroprotective response to RAAS blockade, while a persistent proteinuria is found in those subjects with high dietary sodium intake [62]. The increased glucose uptake in the proximal tubule, due to hyperglycemia, generates an enhanced sodium reabsorption through the SGLT2 channel. Secondary to this, sodium delivery to the distal tubule is reduced and sensed as ineffective circulating volume. This situation triggers a series of mechanisms that lead to hemodynamic changes at the glomerular level, resulting in podocyte injury [63].

### 2.6. Metabolic Acidosis

A recent meta-analysis including 14 trials and 1394 participants [64] found that treating metabolic acidosis with oral alkaline supplementation, or reducing dietary acid intake, increases serum bicarbonate levels (mean difference 3.33 mEq/L, 95% CI, 2.37 to 4.29) and results in a slower decline in eGFR (13 studies, 1329 patients, mean difference –3.28 mL/min/1.73 m^2^, 95% CI, 24.42 to 22.14), along with a reduced risk of progression to KF (relative risk, 0.32; 95% CI, 0.18 to 0.56).

## 3. Prediction of Cardiovascular Events and Progression to Kidney Failure in Patients with DKD and Severely Reduced GFR

Efforts to prevent and treat DKD progression factors with the available measures have not reduced the morbidity and mortality of these patients to the desired level.

Different epidemiological studies have revealed the heterogenicity of DKD features within different rates of CKD progression. Between 19% and 31% of patients may have a non-linear GFR decline [65]. In addition, beyond the classic albuminuric presentation, a non-proteinuric phenotype is described as representing up to 40% of DKD in type 2 diabetics with eGFR <60 mL/min/1.73 m^2^. Typically, in these situations the histological injury is at the vascular and interstitial compartments rather than in the glomerular area. This presentation does not seem to depend on glycemic control and is associated with higher CV risk [66].

The use of predictive models of KF such as the kidney failure risk equation (KFRE), or those associated with the histological classification of diabetic nephropathy score (D-Score) can be useful. However, in DKD these tools present a lower predictive value than in the general cohort of CKD (c-statistic: 0.80 (0.74–0.86)) [67].

In the prediction of CV events, screening for coronary disease in asymptomatic patients is not indicated. Nonetheless, the indication for coronary arteriography should not be restricted, when necessary, despite the risk of kidney injury exacerbation. A carotid and/or femoral ultrasound should be considered to assess the presence of atheromatous disease in patients without cardiovascular disease, as it predicts CV events [3,68,69].

The prediction of CV events and CKD progression can be interfered with by different factors. The incorporation of new drugs to CKD G4, which go beyond glycemic control, with a double objective (cardio and nephro-protection), opens up hope for the future of patients with DKD, in whom the risk of suffering from any of the two events is very high.

## 4. Patient Decision-Making

The transition from CKD G4 to KF (G5) represents a vulnerable period for the patient with elevated risk of adverse events and multiple physiologic and psychosocial changes. It is also a period in which nephrologists must make decisions together with the patient and family members [70]. In patients with diabetes, it may be even more complex because there are usually many comorbidities. Also, there are substantial variations in how this transition is run in different units [71].

Patient education and involvement make this transition easier. A meta-analysis found that self-management support interventions may improve self-management activities [72]. These interventions help patients to better understand KF, compare available treatments and share information with family members [73]. They can be delivered face-to-face as one-to-one or group-based programs or via digital platforms by members of health care teams. The best approach is to tailor the intervention to the individual’s preferences [74]. Few studies have evaluated the utility of self-management education in patients with DKD, but systematic reviews in the population with diabetes have shown a long-term reduction of clinical risk factors [75].

Effective education should cover the necessary information for the patient and family members to answer the following questions [76]: 1. Should I go for KRT or not? 2. If the answer is KRT, dialysis and/or transplantation? 3. Dialysis at home or in-center? 4. If dialysis at home, peritoneal dialysis or home hemodialysis? 5. If transplantation is desired, discuss the question if deceased or living donor?

When kidney function is failing, discussion of the different KRT modalities and selection of a specific therapy should be started promptly [77]. There is not a specific eGFR value for initiating dialysis; rather, it is necessary to offer individualized care. As will be discussed below, a reasonable effort must be made to avoid tunneled catheters as primary vascular access. As life expectancy in some patients is low, persisting efforts to create a vascular access might cause a substantial decrease in their quality of life [30].

If there are no contraindications for KRT (patient candidacy or eligibility profile), it is not clear in subjects with diabetes whether the KRT modality (different modalities of hemodialysis (HD) or peritoneal dialysis (PD)) selected as first-choice has a major impact on outcomes, metabolic profile, diabetes complications and technique survival of the KRT. Therefore, the patient’s preference in selecting KRT should be the driving force for kidney replacement modality. Obviously, patients should be provided with unbiased information [30]. Education on the different options of transplantation and their expected outcomes for patients with diabetes is recommended. Also, information about combined kidney-pancreas transplantation must be included for patients with T1DM and some with T2DM [78,79].

Physicians have the responsibility for initiating and guiding through the advance care planning process. Information needs to be personalized, integrating how their medical and interventions would affect their life and relationships most [80].

## 5. Kidney and Kidney-Pancreas Transplantation in Patients with DKD

Kidney transplantation (KT) is the treatment of choice for the diabetic patient with KFRT, as it increases quality of life and prolongs long-term patient survival [81]. In spite of the increased CV risk in the diabetic patient compared to non-diabetic patients, this improvement in long-term results can even be greater in patients with diabetes as a consequence of the poor results of patients remaining on dialysis [82]. According to Wolfe et al. [81], KT can be associated with a mean increase in life expectancy of 11 years, when compared to the diabetic patient remaining on the waiting list. This survival benefit is mainly due to the significant reduction in CV risk associated with KT, compared to remaining on dialysis [83].

The timing for KT is a relevant aspect to consider. Many studies demonstrate that transplantation in the predialysis phase (preemptive transplantation) is associated with an increased patient survival, both in diabetic and non-diabetic patients [79,84,85,86]. Even more, some studies suggest an increased graft survival [87], although others do not find differences, especially in deceased donor KT [88]. For these reasons, an early referral to the nephrologist for a KT evaluation with enough time for preemptive transplantation is crucial [89]. Some authors recommend referral to the transplant team when eGFR drops below 30 mL/min/1.73 m^2^ [84] in order to complete evaluation and include the patient in the waiting list when it is below 20 mL/min/1.73 m^2^. Living donor KT has the added advantage of the reduced waiting time, thus increasing the possibilities to avoid dialysis initiation [90].

Combined kidney-pancreas transplantation in patients with T1DM and KF brings an additional benefit associated with the normalization of glucose metabolism, the slowing down of progression of organ damage induced by diabetes (mainly retinopathy, neuropathy, nephropathy and cardiovascular damage), as well as a significant improvement in quality of life [78].

Patients with diabetes have an increased surgical risk related to the severity of CV disease. As a consequence, they have a reduced survival rate compared with patients without diabetes [82,91]. This must be taken into account to individualize each patient, considering the risk/benefit ratio of the kidney and/or kidney-pancreas transplantation. A careful evaluation should be done to discard asymptomatic disease (mainly cardiovascular) that might add an excessive risk for surgery and immunosuppression.

## 6. Peritoneal Dialysis in Patients with DKD

In 2001, the NKF-KDOQI guidelines published their recommendations, indications and contraindications of peritoneal dialysis (PD) [92] (Table 3 a,b). In the last decade, for ERBP 2010 [93], KDIGO 2012 [1] and NICE [94], the only factors considered for eligibility in PD were: not having absolute contraindications, having an intact peritoneal membrane, and to have freely chosen this option [95]. There are two modalities of PD, continuous ambulatory peritoneal dialysis (CAPD) and automated peritoneal dialysis (APD), and both can be performed by diabetic patients.

As peritoneal dialysis fluid contains glucose, it must be a factor to take into account when considering PD for diabetic patients. Glycemic control is essential, and the levels of glucose, hemoglobin, and the time of interaction between both, influence the changes in HbA1c. Poor glycemic control appears to be associated with increased morbidity and mortality [96,97,98,99]; however, the relationship between HbA1c levels and morbidity and mortality is controversial. On the one hand, its association is related with higher mortality rates [96], but on the other hand, it is known that there is no association in the first two years of PD [100].

In 2012, a meta-analysis [101] concluded that the use of intraperitoneal (IP) insulin provided adequate glycemic control, which appears to be superior to that observed with subcutaneous (SC) insulin. However, alterations in plasma lipids by IP administration possibly contribute to an increased CV risk. In this regard, no references have been found in PUBMED in recent years.

As compared with conventional hemodialysis, PD is associated with a slower decrease in residual kidney function (RKF) [102]. An observational study showed that glycemic control was not associated with changes in RKF during the first year on PD [103]. However, the slope of loss of RKF is more pronounced in patients with diabetes, especially in the first year, diminishing in the following year and, starting from a higher initial urine production, they also experience a greater decrease in it [104].

The associated multi-comorbidity that affects diabetic and non-diabetic patients alike may be present from the beginning of treatment, and the numerous barriers that may arise over time (Table 4) must be overcome by complying with international evidence-based clinical guidelines, with measures that contribute to reduce the failure of the technique, as well as by providing care, psychological advice, and assistance for dialysis at home [105,106,107,108,109,110,111,112,113].

## 7. In-Center Hemodialysis in Patients with DKD. Vascular Access Problems in Patients with Diabetes

There are no specific contraindications for performing HD in patients with DKD. However, patients with diabetes may present important complications to be taken into account. These complications are related to atherosclerosis and cardiomyopathy, which cause most morbidity and mortality, as well as problems with the vascular access [114]. The excessive cardiac morbidity and mortality of patients with diabetes seem to be mediated via ischemic disease rather than progression of cardiomyopathy while on dialysis therapy. Foley et al. found that only 16% of 432 patients with diabetes at the start of dialysis had normal left ventricular size and systolic function [115]. These cardiological problems condition tolerance to HD sessions, favoring intradialytic hypotension [116], which is the most frequent complication of the HD technique.

In 1972, Chavanian wrote: “…there is little prospect of improving the quality of life for patients with diabetic nephropathy and renal failure, and survival is likely to be short. Dialysis for such patients may be considered as a palliative measure with little prospect of long-term survival” [117]. Since then, fortunately, things have improved a lot, but the mortality of patients with diabetes on HD continues to be elevated [118,119].

There is a series of specific problems in patients with diabetes, such as intradialytic hypotension [116], anemia, altered parathormone activity etc. Among them, we want to highlight glycemic control [120] and vascular access-related problems [121,122].

In patients with diabetes on hemodialysis, blood glucose levels are associated not only with factors related to KF but also with HD. In patients on HD, predialysis glucose levels are used instead of fasting glucose. Glycosylated hemoglobin (HbA1c) values tend to be lower in HD patients, indicating a glycemic control apparently better than what they actually have [123,124]. There are several reasons for this, such as that the life span of erythrocytes is shortened, the blood loss occurring during HD therapy, and the use of erythropoiesis-stimulating agents (ESAs). For these reasons, some guidelines recommend using glycated albumin (GA) [125]. Also, glycemic control in patients on HD differs widely between non-dialysis and dialysis days. HD can also induce hyperglycemia. As glucose diffuses from plasma to dialysis fluid, glucose levels decrease during the dialysis session and increase after HD. However, also, plasma insulin is removed during the HD session due to its adsorption on the dialyzer membrane. According to these conditions, both HD-induced hypoglycemia and HD-associated hyperglycemia can occur during and after HD sessions [126].

Since a considerable number of hemodialysis patients are diabetic and elderly, the vascular access issue is one of the major challenges for nephrologists and vascular surgeons. Both the creation and proper development of the vascular access are crucial for adequate dialysis treatment, while ensuring adequate longevity of use, sufficient blood flow to achieve the necessary dialysis dose, and a minimal complication rate [127]. However, the presence of DM is one of the main risk factors for vascular access failure, as it causes atherosclerosis in small and medium-sized vessels, resulting in calcification and arterial stenosis [128]. Calcified atherosclerosis often implies inadequate flow, as the arteries are unable to dilate. Thus, compared with patients without diabetes, incident diabetic patients on HD have worse arteriovenous fistula (AVF) patency rates [121] and a higher risk of AVF failure [122]. A meta-analysis [122], with 23 papers analyzed, which were published between 1998 and 2016, regarding AVF complications in the diabetic population, studied the outcomes in 930 diabetic subjects with AVF. The work revealed that patients with diabetes have a higher rate of AVF failure compared to non-diabetics. The authors argue several reasons for this finding. First, diabetic disease has a high tendency for platelet aggregation and a higher prevalence of thrombotic formations [129]. In part, this is due to the release of bioactive substances as a consequence of hyperglycemia and glycosylation end products, which injure the internal wall of blood vessels. Associated with this, patients with diabetes have a higher prevalence of atherosclerosis, as well as a greater severity of vascular lesions. To these features must be added that the patient with diabetes (mainly type 2) is more exposed to numerous blood extractions, and intravenous infusions and punctures during the course of the disease.

A study by Gołębiowski et al. [130] analyzed 166 patients with DKD who had a new AVF created and analyzed outcomes and complications according to the different types of fistulas created. More than 80% of the patients were T2DM patients. Atherosclerotic changes were observed at the level of the forearm in 60% of diabetic subjects. Among these patients, in about half of them an adequate forearm AVF could be created in the first procedure, while in the other half an additional intervention was necessary. Despite this, in only 10% of all diabetics (*n* = 166) did atherosclerosis pose a significant obstacle to AVF creation, despite requiring additional procedures. Regarding the site of AVF creation, the authors agree that the preferred site is the wrist region; however, in patients with diabetes, due to atherosclerotic changes, the elbow area is frequently used.

Another important study [131] analyzing 347 AVFs and 799 vascular access procedures, comparing diabetic patients with non-diabetic patients, observed that the number of deceased patients, of those who had been switched to peritoneal dialysis, who had undergone surgical closure of the AVF or who had been switched to hemodialysis via indwelling central venous catheter, was higher in patients with diabetes. In addition, probably in relation to vascular complications, non-diabetic subjects had been more frequently kidney transplanted. Also, probably in association with a high comorbidity prior to the creation of the AVF that led to its creation at the elbow, these patients (with localized AVF at the elbow) had a higher mortality and a greater number of active AVF closures. In this study, 74% of diabetics required an AVF at the elbow, compared to 32% in non-diabetic patients. The selection of the type of fistula should take into account the shorter life expectancy of some diabetic patients. In the study just described, at the end of the follow-up period (mean follow-up 31 ± 19.3 months), only 20.5% of diabetic patients were alive with a functioning vascular access in the same extremity, compared to 44.6% of non-diabetic patients. For this reason, the authors suggest creating vascular access in the elbow region if limited survival on hemodialysis is anticipated. This allows a higher dose of dialysis to be administered and a better quality of life than non-maturable peripheral anastomoses. A review by the same author [132] provided data from his experience. Within five years, 748 underwent primary AV fistula construction. The author indicates that 15% of diabetic patients (compared to 2% of non-diabetic patients) had contraindications to the performance of any vascular access due to cardiovascular comorbidities, such as peripheral ischemia or congestive heart failure. Of all the patients, 24% were diabetic. As in the previous study, diabetic patients required the creation of the AVF at the elbow in a higher proportion than non-diabetic patients (76% vs. 38%).

As a recommendation, it is important to bear in mind that a detailed and systematic preoperative study helps in the choice of the anatomical area to be preferred for the first AVF, since adequate maturation of the AVF requires a healthy vascular tree, free of calcifications. If the arteries are calcified, the elbow should be the first choice for AVF creation, as calcification at this level is less pronounced than in the wrist arteries. However, these present an increased risk of steal syndrome, as involvement of arteries distal to the elbow results in increased peripheral resistance, diverting most of the flow into the fistula, leading to hand ischemia [133]. Jennings et al. described that proximal radial artery-based AVFs offer excellent functional patency with a low risk of dialysis access-related steal syndrome [134]. Given the increased risk of infections and the complexity of their treatment in the diabetic patient, AVF should be the vascular access of first choice for most diabetic patients, and mainly if they are elderly patients, so the use of central venous catheters is usually an alternative only when the AVF is to be dispensed with. However, it is a valid option in diabetic patients, as is the switch to peritoneal dialysis.

The EUDIAL group [135] reminds that for proper clinical practice the health care team should strive for the most appropriate vascular access for each patient, based on local experience, patient comorbidities, physical examination, ultrasound mapping and surgical anatomy. It further recommends that practitioners should employ, mainly in the elderly and/or diabetic population, that surgical strategy that minimizes to the maximum the complications of VA, such as AVF failure, steal syndrome and congestive heart failure [136].

## 8. Home Hemodialysis in Patients with DKD

The increased incidence and prevalence of patients with diabetes and CKD G5 who require dialysis is a challenge to optimizing accessibility to the current available kidney replacement therapy (KRT).

Home HD offers the possibility of increasing the frequency, duration of the session, tolerance, and efficacy of substitution treatment since it directly affects not only a series of very prevalent factors in patients with advanced and elderly DKD, such as BP control, ventricular hypertrophy, anemia, mineral metabolism, peripheral vascular disease, autonomic dysfunction, hemodynamic instability and nutrition but also their quality of life, with a favorable impact on survival [137,138,139,140].

International [141] clinical practice guideline for hemodialysis adequacy: 2015 update especially recommend intensive HD regimens, such as those provided by Home HD, precisely for this profile of patients in whom treatment must be individualized, with the least impact on their quality of life.

In patients with diabetes and CKD G5 on home HD, due to its comorbidities, special attention is required to its stability during medical follow-up. Also, is important to avoid major clinical or hemodynamic incidences during HD sessions, to have an adequate vascular access and commitment to self-care, as well as socio-family support that, on occasions, includes a caregiver committed to performing the Home HD and/or assisting it [142].

For patients with DKD, the absolute and relative contraindications for performing home HD [143] are the same as for the rest of the population with CKD who require KRT.

Whilst KRT offers better results, it is our obligation to offer patients with DKD the option of home HD in the informed decision-making process. We must offer them the best adequate dialysis possible, individualized and respecting their autonomy wherever their preferences are located and their lifestyle at the center of choice.

## 9. Patients with DKD Who Could Benefit from Comprehensive Conservative Care

For people with diabetes, KF is a potentially devastating condition, markedly increasing cardiovascular risk and potentially leading to premature death [4]. Patients with DKD have multiple comorbidities, suffer higher overall hospitalization, higher complication rates, and the most truncated life expectancy on dialysis compared to other patient groups of any age [3]. Many of them at the commencement of dialysis are frail, which increases both morbidity and mortality and limits the individual’s ability to cope with dialysis. Functional dependence and impaired intellectual status are also poor prognostic factors [144,145,146,147,148]. Thus, on certain occasions, nephrologists and patients themselves find it difficult to decide whether long-term dialysis is the best choice, especially when patients have multiple comorbidities. A long term of KRT, with the absence of any prospect of resolving the clinical situation in those patients and no alternative to receive a KT, has a negative effect on their quality of life [149]. All these aspects must be evaluated before starting KRT, since survival and quality of life are indeed not necessarily better by undergoing dialysis, especially for elderly patients with diabetes and/ or with many diabetes-related complications.

Comprehensive conservative kidney care is understood as a holistic patient-centered care that not only includes interventions to delay CKD progression and minimize complications but also provides detailed communication, shared decision-making, advance care planning, and psychological and family support; however, it does not include dialysis [150]. Older patients, who are socially isolated and have reduced functional capacity and diabetes, are more likely to be offered comprehensive conservative care rather than dialysis [151]. There is almost no evidence about which models of care and which interventions might be most beneficial in the population with diabetes. Many studies have compared outcomes of CKD G5 under conservative versus KRT in elderly patients. Age and DM are identified as decisive factors on prognosis in most of them [149,152]. However, there is a lack of comparative studies focused on the population with diabetes. A case report on a middle-aged woman with DKD that had been followed for 15 years in CKD G5 despite severe disease highlights the need for further studies, not limited exclusively to elderly patients, to verify the efficacy of non-dialysis treatment in patients with diabetes [152].

Knowledge of the main etiologies of CKD and the metabolic alterations and associated symptoms is an important element in providing patients with good palliative care. It should be noted that excessive decreases in blood pressure and glucose levels may be linked with severe complications in CKD G5. Approaches to managing diabetes vary. Individuals previously on insulin therapy are often required to decrease dosages, since insulin is metabolized in the renal tubular cells and has its half-life prolonged in CKD G5 [153]. A recent study, among a cohort of US veterans, shows the need for different glycemic strategies based on whether there are plans to transition to dialysis versus pursuing comprehensive conservative care among DKD patients by relating mortality to different glycated hemoglobin targets in each of the groups [154].

In the absence of robust evidence in favor of comprehensive conservative care in patients with diabetes, nephrologists may consider this option in older patients or in those who have higher levels of comorbidity and poorer functional status, independent of the age.

## 10. Conclusions

When patients with DKD progress to severely reduced GFR (CKD G4), they are in a vulnerable situation in which it will be necessary to continue efforts to delay the progression to KF (CKD G5), while controlling CV risk factors and/or vascular complications associated with diabetes. It should also be borne in mind that in the situation of advanced kidney injury, less scientific evidence is available and the use of certain pharmacological tools is restricted. At the same time, it is important to recall that when the progression of kidney disease cannot be delayed, other possibilities to improve the patient’s quality of life, such as dialysis, renal transplantation or conservative management, should be explored. Therefore, a shared decision-making process between health care agents, patients and family members should be initiated at this point. In this process, the clinical circumstances and preferences of the patient should be taken into account. In certain cases, it should be considered whether comprehensive conservative care is appropriate. For all this, it is essential to have an adequate multidisciplinary team and sufficient time to devote to the patient and his or her family.

## Figures and Tables

**Figure 1 jcm-10-02943-f001:**
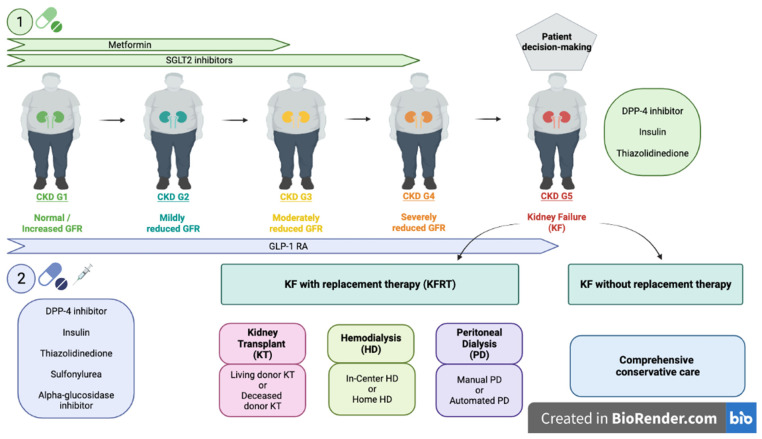
DKD progression and management recommendations using nomenclature recommended by Executive Summary and Glossary from a Kidney Disease: Improving Global Outcomes (KDIGO) Consensus Conference ① and KDIGO Diabetes Management in CKD Guideline ②.

**Table 1 jcm-10-02943-t001:** CKD progression risk factors in DKD.

Modifiable	Non-Modifiable
Socioeconomic factors	Duration of diabetes
Hyperglycemia	Familial and genetic factors
Hypertension	Autonomic neuropathy
Lipids	
Diet	
Correcting bicarbonate	
Obesity	
Drug and procedure toxicity	
Urinary infections	
Anemia	

**Table 2 jcm-10-02943-t002:** Usual and renal function-adjusted doses of Metformin, SGLT2 inhibitors and GLP-1 RA.

Drug	Dose	CKD Adjustment
**Metformin**		
Immediate release	Initial 500–850 mg once daily Titrate upwards by 500–850 mg/d every seven days until maximum dose	For an eGFR between 45–59 mL/min/ 1.73 m^2^, dose reduction if risk of hypoperfusion and/or hypoxemia. Halve the dose if eGFR 30–45 mL/min/1.73 m^2^ Discontinue if eGFR < 30 mL/min/1.73 m^2^
Extended release	Initial 500 mg daily Titrate upwards by 500 mg/d every seven days until maximum dose	
**SGLT2 inhibitors**		
Empagliflozin	10–25 mg once daily	No dose adjustment if eGFR ≥ 45 mL/min/1.73 m^2^ Discontinue if eGFR persistently < 45 mL/min/1.73 m^2^
Canagliflozin	100–300 mg once daily	No dose adjustment if eGFR > 60 mL/min/1.73 m^2^ 100 mg daily if eGFR 30–59 mL/min/1.73 m^2^ Avoid initiation with eGFR < 30 mL/min/1.73 m^2^ Discontinue when initiating dialysis
Dapagliflozin	5–10 mg once daily	No dose adjustment if eGFR ≥ 45 mL/min/1.73 m^2^ Not recommended if eGFR < 45 mL/min/1.73 m^2^ Contraindicated with eGFR < 30 mL/min/1.73 m^2^
**GLP-1 RA**		
Dulaglutide	0.75 mg and 1.5 mg once weekly	No dosage adjustment Use with eGFR > 15 mL/min/1.73 m^2^
Exenatide	10 μg twice daily	Use with CrCl > 30 mL/min
Exenatide (Extended-release)	2 mg once weekly	
Liraglutide	0.6 mg, 1.2 mg, and 1.8 mg once daily	No dosage adjustment Limited data for severe CKD
Lixisenatide	10 μg and 20 μg once daily	No dosage adjustment Limited data for severe CKD
Semaglutide (injection)	0.5 mg and 1 mg once weekly	No dosage adjustment Limited data for severe CKD
Semaglutide (oral)	3 mg, 7 mg, or 14 mg daily	

Abbreviations: CKD: chronic kidney disease; CrCl, creatinine clearance; eGFR, estimated glomerular filtration rate; SGLT2, sodium-glucose cotransporter-2; GLP-1 RA, glucagon-like peptide-1 receptor agonists. For SGLT2 inhibitors, the adjusted doses correspond to the indications approved by the U.S. Food and Drug Administration (FDA).

**Table 3 jcm-10-02943-t003:** Contraindications of PD: (a) according to the clinical practice recommendations of the NKF-KDOQI (Golper TA, Am J Kidney Disease 2001); (b) other contraindications mentioned in the literature.

**(a)**	
**Absolute**	**Relative**
Documented loss of peritoneal function or extensive abdominal adhesions limiting dialysate flow.	Recent intra-abdominal bodies (e.g., less than 4 months, vascular prosthesis, ventriculopertioneal shunt).
Patients who have physical or mental disabilities in the absence of a suitable assistant.	Peritoneal leaks.
Patients with uncorrectable mechanical defects that prevent effective PD or increase the risk of infection:	Limitations due to body size.
Surgically irreparable hernia.	Intolerance to the volumes of PD necessary to reach the adequate dialysis dose.
Omphalocele.	Active inflammatory or ischemic bowel disease.
Gastroschisis.	Abdominal wall or skin infection.
Diaphragm hernia.	Morbid obesity (in small individuals).
Bladder exstrophy.	Severe malnutrition.
	Frequent episodes of diverticulitis.
**(b)**	
**Absolute**	**Relative**
*Medical:*	*Medical:*
Large abdominal aortic aneurysm.	Abdominal adhesions.
Previous major abdominal surgery and with large abdominal scars.	Planned abdominal surgery.
Severe lung disease with poor lung function.	Ostomies, colostomy, ileostomy.
	Gastric tube.
	Severe gastroparesis.
	Polycystic kidney disease (very large kidneys).
	Intestinal cancer.
	Diseases that limit manual dexterity or upper limb amputations.
	Presence of ventriculoperitoneal communication.
*Social:*	*Social:*
The place of residence does not allow it.	Inability to perform the technique with asepsis and no family support.
Work or employment does not allow it.	

**Table 4 jcm-10-02943-t004:** Barriers that may arise over time patients with PD.

Aging.
Frailty.
Overweight.
Loss of autonomy to self-care.
Visual, auditory and cognitive problems.
Psychosocial and social problems such as isolation.
Burnout of the patient or caregiver.
Inflammation and malnutrition.
Loss of residual kidney function (RKF).
Volume overload and cardiovascular disease (CVD).
Stroke and peripheral arterial disease (PVD).
Metabolic syndrome secondary to glucose absorption.
Alterations in the functionality of the peritoneal membrane.

## Data Availability

Not applicable.

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
