# Peer review of "Management of Kidney Failure in Patients with Diabetes Mellitus: What Are the Best Options?"

_jcm, 2021, doi:10.3390/jcm10132943_

Round 1
Reviewer 1 Report
I read again the manuscript entitled “Kidney Failure in Patients with Diabetes. What Is the Best Option?” Juan M. Buades et al, after authors introduced modifications according to previous suggestions to publish in International Journal of Clinical Medicine
I confirm my personal feeling that the manuscript is well redacted and focused, but it does not introduce any noticeable new concept for the readers. It appears as a book chapter of medicine or nephrology.
Apart, the introduction has some minor concerns as shown below which must be corrected. The whole manuscript should be read by a senior English medical writer to improve English style, not the grammar.
65 important instead of import
65 Rewrite: Kidney replacement therapy (KRT) refers to either un- 65 dergoing kidney transplantation or initiating dialysis Therapy
65 Rewrite and cite guideline: For the former, there are 66 clear guidelines which aids to clarify those patients who qualify for transplant and when 67 it is the right moment to perform it
68 Rewrite: However, defining the appropriate time to initiate 68 kidney dialysis is a challenging task
75-77: when patients present with severely re- 75 duced GFR (stage G4), they are evaluated by multidisciplinary teams led by nephrolo- 76 gists. The importance of this approach is to try to predict which patients have a greater 77 risk of mortality and which are more likely to progress to KF If patients have severely reduced GFR they do not need which are more likely to progress. They are already in severe KF
83 Rewrite: this takes on a further difficulty due 83 to
Author Response
Reviewer #1 Comments:
I read again the manuscript entitled “Kidney Failure in Patients with Diabetes. What Is the Best Option?” Juan M. Buades et al, after authors introduced modifications according to previous suggestions to publish in International Journal of Clinical Medicine
I confirm my personal feeling that the manuscript is well redacted and focused, but it does not introduce any noticeable new concept for the readers. It appears as a book chapter of medicine or nephrology.
Apart, the introduction has some minor concerns as shown below which must be corrected. The whole manuscript should be read by a senior English medical writer to improve English style, not the grammar.
65 important instead of import
We thank the reviewer for the suggestions and review. We have corrected the error.
65 Rewrite: Kidney replacement therapy (KRT) refers to either undergoing kidney transplantation or initiating dialysis Therapy
We have rewritten this sentence to facilitate its comprehension (lines 68-72).
65 Rewrite and cite guideline: For the former, there are clear guidelines which aids to clarify those patients who qualify for transplant and when it is the right moment to perform it
We have rewritten the text and added the corresponding citation from the guideline (lines 73-75).
68 Rewrite: However, defining the appropriate time to initiate kidney dialysis is a challenging task
We have rewritten this sentence to facilitate its comprehension (lines 75-77).
75-77: when patients present with severely reduced GFR (stage G4), they are evaluated by multidisciplinary teams led by nephrologists. The importance of this approach is to try to predict which patients have a greater risk of mortality and which are more likely to progress to KF. If patients have severely reduced GFR they do not need which are more likely to progress. They are already in severe KF
When referring to patients with “severely reduced GFR”, we rely on the nomenclature report recommended by the Improving Global Outcomes (KDIGO) Consensus Conference (2020) in which this term (severely reduced GFR) is used for patients with GFR 29-15 ml/min/1.73m2. These patients may progress to a more advanced stage of kidney disease (G5) where the GFR decreases to less than 15 mL/min/1.73m2 (Kidney Int. 2020 Jun;97(6):1117-1129. doi: 10.1016/j.kint.2020.02.010).
83 Rewrite: this takes on a further difficulty due to
We have rewritten this sentence to facilitate its comprehension (lines 90-92).

Reviewer 2 Report
Dr Buades and Colleagues have done a great job refining the manuscript. Most of the changes have been incorporated and overall the flow is improved, with a finer focus on the topic at hand.
I have few comments/ suggestions for the revised manuscript.
- Abstract: point number 5: it can also be stated "if transplantation is desired, discuss the options"
- In the section: combining the different risk factors appear more organized. Great Idea by Authors here.
"Duration, socioeconomic factors and familial and genetic factor"
I would suggest to either remove this statement below or reference it, I am not sure if the Author's opinion here would be true.
"Poor Dietary habits can also be at the other end of the socioeconomic spectrum, so we should look for citable references to this statement or change it. Similar for physical activity as well"
3. Page 4 Line 187 "About referring to inheritance patterns etc it may be
too little on a big topic, but if authors would like to keep it then it should be referenced"
4.Page 5 Line 223-225 needs to be referenced.
5. Quoting DOPPS Dialyis Outcomes and
DOPPS is a network that has several studies, would reference this like "Review of DOPPS Study by xyz et al showed that" instead of saying DOPPS study.
6. May add from CREDENCE AND DAPA CKD Study as SGLT-2 inhibitors have shown robust data in slowing CKD progression, would encourage, as they can potentially add 2-3 sentences more here if desired.
7. I would suggest using Vascular Access in place of VA.
8.Page 13 Line 574: "mortality continues to be higher" it would be better to explain mortality is higher than "compared to what group".
9. Page 13 Line 579 "we use" does that imply that the authors in their practice use it?
10. Page 14 Line 625: this statement is very long without any commas, "needs to be paraphrased from a grammatical standpoint, or broken down into two sentences". Its so long it is hard to get what it is trying to compare and highlight.
11. Conclusions are much refined. I may suggest that
"in addition to delaying progression of kidney failure there is a need to appropriately diagnose, and manage a variety of complications including Cardiovascular......" or something similar to emphasize that when progression cannot be slowed, other options need to be explored like Dialysis, Transplant or Conservative Kidney Management.
12. Lastly this manuscript would benefit from another figure or flowsheet to help summarize the keys messages of this review.
13. New sections added in Introduction, and vascular access are very nicely done.
Author Response
Reviewer #2 Comments:
Dr Buades and Colleagues have done a great job refining the manuscript. Most of the changes have been incorporated and overall the flow is improved, with a finer focus on the topic at hand.
I have few comments/ suggestions for the revised manuscript.
Abstract: point number 5: it can also be stated "if transplantation is desired, discuss the options"
We thank the reviewer and agree with the suggestion (line 42).
In the section: combining the different risk factors appear more organized. Great Idea by Authors here. "Duration, socioeconomic factors and familial and genetic factor". I would suggest to either remove this statement below or reference it, I am not sure if the Author's opinion here would be true. "Poor Dietary habits can also be at the other end of the socioeconomic spectrum, so we should look for citable references to this statement or change it. Similar for physical activity as well"
We fully agree with the reviewer's comment and suggestion. We agree that poor dietary habits and low physical activity may also occur in subjects with a high economic status. For this reason, we have removed this statement. Thank you for highlighting it. (line 178).
- Page 4 Line 187 "About referring to inheritance patterns etc it may be too little on a big topic, but if authors would like to keep it then it should be referenced"
We agree on the importance of this topic. We have added bibliographic citations to allow further consideration of this issue (lines 182-187).
4.Page 5 Line 223-225 needs to be referenced.
We have referenced these statements.
- Quoting DOPPS Dialysis Outcomes and DOPPS is a network that has several studies, would reference this like "Review of DOPPS Study by xyz et al showed that" instead of saying DOPPS study.
We agree with the reviewer's suggestion and have changed the statement (line 197).
- May add from CREDENCE AND DAPA CKD Study as SGLT-2 inhibitors have shown robust data in slowing CKD progression, would encourage, as they can potentially add 2-3 sentences more here if desired.
We agree with the reviewer and have added information on CREDENCE and DAPA-CKD clinical trials (lines 218-233).
- I would suggest using Vascular Access in place of VA.
We have replaced “VA” by “vascular access”.
8.Page 13 Line 574: "mortality continues to be higher" it would be better to explain mortality is higher than "compared to what group".
We have rewritten this sentence to facilitate its comprehension (line 739).
- Page 13 Line 579 "we use" does that imply that the authors in their practice use it?
We have replaced the term by: In patients on HD, predialysis glucose levels “are used” instead of fasting glucose (line 744).
- Page 14 Line 625: this statement is very long without any commas, "needs to be paraphrased from a grammatical standpoint, or broken down into two sentences". Its so long it is hard to get what it is trying to compare and highlight.
We have rewritten this sentence, breaking it down into four sentences to facilitate its comprehension (lines 928-942).
- Conclusions are much refined. I may suggest that "in addition to delaying progression of kidney failure there is a need to appropriately diagnose, and manage a variety of complications including Cardiovascular......" or something similar to emphasize that when progression cannot be slowed, other options need to be explored like Dialysis, Transplant or Conservative Kidney Management.
We agree with the reviewer and have completed the conclusion as suggested (lines 928-943).
- Lastly this manuscript would benefit from another figure or flowsheet to help summarize the keys messages of this review.
A flow chart has been added to summarize the keys messages of this review (Figure 1).
- New sections added in Introduction, and vascular access are very nicely done.
We thank the reviewer for the very valuable comments and suggestions.

Reviewer 3 Report
Diabetes mellitus (DM) is a leading cause of chronic kidney disease (CKD), in particular end-stage renal disease (ESRD), especially in western countries, affected by an epidemic of obesity. After analysis the data, provided by US Renal Data System (USRDS), it might be expected that the number patients one hand requiring any type of renal replacement therapy (RRT) and on the other suffering from cardiovascular consequences of DM, would grow continuously. There is a crucial issue to identify those individuals at the early stages of CKD and provide tailored care. Therefore, I find the paper by Buades and co-authors as of utmost importance and interest.
After reading above mentioned work I have some remarks, which should be clarified, before further processing.
- “Chronic kidney disease (CKD), which is defined by persistent reduction in glomerular filtration rate (GFR) and/or the presence of other signs of kidney damage for more than 3-months, is classified according to the categories of GFR and albuminuria…”
Please reconsider this part. In my opinion CKD is defined by either reduction in GFR or presence of other sings of kidney damage. I understand your idea but in this form the sentence suggests that GFR reduction is essential to define CKD.
- “It is widely accepted that the GFR level alone should not be used as a reason to start KRT, and that signs and symptoms associated with kidney failure (KF) should be considered…”
Authors do not specify what do they mean as kidney failure (KF). Does this term refer to CDK Stage 5, known also as end-stage renal disease (ESRD)? It should be specified.
- “…canagliflozin treatment reduced the risk of KF in participants with eGFR ≥30 ml/min/1.73…”
The same issue as in the previous point. Please clarify whether KF is the same as CKD Stage 5.
- “severely reduced GFR”
I would advise not to use a term like above. As nephrologist I somehow feel what is your idea but please be more precise.
- “When patients with DKD progress to severely reduced GFR (CKD G4-G5)”T
The same comment as in 4.
Author Response
Reviewer #3 Comments:
Diabetes mellitus (DM) is a leading cause of chronic kidney disease (CKD), in particular end-stage renal disease (ESRD), especially in western countries, affected by an epidemic of obesity. After analysis the data, provided by US Renal Data System (USRDS), it might be expected that the number patients one hand requiring any type of renal replacement therapy (RRT) and on the other suffering from cardiovascular consequences of DM, would grow continuously. There is a crucial issue to identify those individuals at the early stages of CKD and provide tailored care. Therefore, I find the paper by Buades and co-authors as of outmost importance and interest.
After reading above mentioned work I have some remarks, which should be clarified, before further processing.
- “Chronic kidney disease (CKD), which is defined by persistent reduction in glomerular filtration rate (GFR) and/or the presence of other signs of kidney damage for more than 3-months, is classified according to the categories of GFR and albuminuria…”
Please reconsider this part. In my opinion CKD is defined by either reduction in GFR or presence of other sings of kidney damage. I understand your idea but in this form the sentence suggests that GFR reduction is essential to define CKD.
We are very grateful to the reviewer for the suggestions and comments. According to KDIGO 2012 we have defined CKD as the presence of markers of kidney damage or a glomerular filtration rate (GFR) <60 mL/min/1.73 m2, for 3 months or more, irrespective of cause (Group KDIGO, “KDIGO 2012 clinical practice guideline for the evaluation and management of chronic kidney disease Clinical Practice Guidelines,” Kidney Int Suppl., vol. 3, no. 1, pp. 1-150., 2013). (lines 55-57).
- “It is widely accepted that the GFR level alone should not be used as a reason to start KRT, and that signs and symptoms associated with kidney failure (KF) should be considered…”
Authors do not specify what do they mean as kidney failure (KF). Does this term refer to CDK Stage 5, known also as end-stage renal disease (ESRD)? It should be specified.
We have explained the terminology in the text (lines 68-72). We refer to kidney failure (KF) as a GFR <15 mL/min per 1.73 m2 or treatment by dialysis (G5), as it has been recently suggested by the KDIGO Consensus Conference (Nomenclature for kidney function and disease: report of a Kidney Disease: Improving Global Outcomes (KDIGO) Consensus Conference, Kidney International (2020) 97, 1117–1129).
- “…canagliflozin treatment reduced the risk of KF in participants with eGFR ≥30 ml/min/1.73…”
The same issue as in the previous point. Please clarify whether KF is the same as CKD Stage 5.
We have clarified the statement (line 69) following the same suggestions described above (Nomenclature for kidney function and disease: report of a Kidney Disease: Improving Global Outcomes (KDIGO) Consensus Conference, Kidney International (2020) 97, 1117–1129).
- “severely reduced GFR”
I would advise not to use a term like above. As nephrologist I somehow feel what is your idea but please be more precise.
Following the suggestions of the KDIGO experts (Nomenclature for kidney function and disease: report of a Kidney Disease: Improving Global Outcomes (KDIGO) Consensus Conference, Kidney International (2020) 97, 1117–1129), by “severely reduced GFR” we refer to a glomerular filtration rate between 15 and 29 mL/min per 1.73 m2 (G4).
- “When patients with DKD progress to severely reduced GFR (CKD G4-G5)”
The same comment as in 4.
We thank the reviewer for the observation. We would like to refer to the response above. However, in this case we have removed “G5” from the sentence because it was mistakenly included (line 926).

Reviewer 4 Report
The paper is comprehensive and contain a lot of practical issues, for non-academic nephrologist or GPs. However the are some issues to be raised:
KDIGO - the newest version contain treatments schemes (they should be included to the paper), especially newest info about connection of SGLT2 and GLP1
The fragment about vascular access should be re-write – there is a lot of paper concerning vascular access in diabetic patients.
More data about newest drugs with more detailed treatment standards should be included .
Author Response
Reviewer #4 Comments:
The paper is comprehensive and contain a lot of practical issues, for non-academic nephrologist or GPs. However the are some issues to be raised:
KDIGO - the newest version contain treatments schemes (they should be included to the paper), especially newest info about connection of SGLT2 and GLP1
We thank the reviewer for the suggestion, which seemed to be very appropriate. We have complemented the text by providing information on the therapeutic schemes recommended by the international KDIGO guidelines. We have also added the Figure 1 and the Table 2 to assist in the reading of the recommendations. (lines 316-344).
The fragment about vascular access should be re-write – there is a lot of paper concerning vascular access in diabetic patients.
The authors agree with the reviewer's comment. In fact, after the first review we have rewritten the entire section on vascular access in diabetic patients, following a wide examination of the abundant literature available. However, due to the extensive length of this review and the planned multidisciplinary approach, we have tried to synthesize the information to avoid overwhelming the reader.
More data about newest drugs with more detailed treatment standards should be included .
We have added in the section on management of hyperglycemia some of the characteristics that favor the use of the new hypoglycemic agents. We have also included the Figure 1 to facilitate the understanding of when they can be used; and the Table 2 to summarize the dosage of the more important antiglycemic medications. Thanks to the reviewer for this very pertinent suggestion. (lines 318-418)
Again, we would like to thank the reviewers for their arduous work and valuable suggestions. As you will see in the version with marked tracked changes, we have made an effort to change practically the entire focus of the article, using the recommendations received. We believe that the effort is worth it in view of the possibility of having the honor of publishing this review in the Journal of Clinical Medicine.
Thank you again for your suggestions and collaboration.

This manuscript is a resubmission of an earlier submission. The following is a list of the peer review reports and author responses from that submission.
Round 1
Reviewer 1 Report
In this review article Dr Buades and Colleagues have aimed at reviewing Diabetic Kidney Disease (DKD), Risk factors for Progression of Chronic Kidney Disease (CKD) and Management Options for both complications and long term therapy including Transplant, Dialysis Therapy options and conservative kidney care.
Overall Comment: Authors have chosen a very important and widely worked upon topic for clinical review. They have worked hard and did a great job to compile all the important aspects that may be encountered when dealing with patients of Diabetic Kidney Disease (DKD). However it appears that their scope of the topic at hand, significantly expanded beyond what could be reviewed in this space. There is a lack of focus on specific topics, instead an attempt is made to include as many topics as able, without really delving into them in detail: which should be one of the aims of a review article. Instead of reviewing latest evidence and then summarizing it for the reader they have in most places simply stated evidence for or against an argument or observation without mentioning what would be a summary to guide readers. It appears to start from the Title and page 2/19, Paragraph 2 Line 13 that where Authors have stated a number of issues e.g. late nephrology referrals, then stating “It is important to be able to predict which patients are at the highest risk of mortality and which ones are more likely to progress to KF, as they will be the ones who will really benefit from this multidisciplinary management”.
I would suggest clearly specifying in 2-4 sentences exactly what the authors aim to review in this article (any idea could be placing the 10 headings into 3 or 4 and then stating them in introduction). If the aim is to show how care of Advanced CKD due to Diabetes is different than Non Diabetic CKD, then appropriate comparisons should be made in each paragraph e.g. in each of the points in the section “Slowing diabetic kidney disease progression to G5 in patients with G4 CKD” instead of simply stating what happens in DKD only. Also it would help to limit the focus of the review to Diabetics who have been classified historically as Type II diabetics, as the material covered in the review is mainly related to Type II diabetics and only a few places specifics of Type I DM have been described which is not enough for a review article. It may also increase the focus on DKD and would allow space to incorporate all the suggestions mentioned above and in subsequent paragraphs.
I suggest removing the Kidney Pancreas Transplantation section as well as Islet Cell or Stem Cell Transplantation sections.
Strengths: Important topic that will be very helpful for everyone involved in the care of patients (as mentioned by the authors e.g. endocrinologists, nurses, vascular surgeons, radiologists etc) with Advanced Kidney Disease. More than required areas have been identified for review which can easily be reduced to make it more refined, focused and in depth.
Weakness: Lacks focus, and in depth review of most important topics like Vascular access, HTN Management in Diabetics AND how they are different compared to Non Diabetic Patients who have CKD. Instead of a detailed review authors have tried to superficially touch a lot of topics.
Specific Feedback with potential suggestions:
Page 2:
“Instead of "on the other hand" would suggest presenting KFRT as a morbidity or complication from DKD”
“Improvement of prognosis depends on the candidate getting KFRT and by improving it authors want to state that as compared to not getting KFRT?”
“There are large variations in the incident rates of KRT” do authors mean among DKD patients?
When discussing “ In general, it is widely accepted that the GFR level alone should not be used as a reason for starting KRT, and that the signs and symptoms associated with KF should be considered”
Can probably remove Transplantation here and simply focus the timing question on start of dialysis Therapy, in which case then it will be true. For transplantation there are much clearer guidelines, GFR cut offs and pre-transplant evaluations which have clarified who qualifies for transplant and when. Dialysis on the other hand has no clear rules for obvious reasons such as functional status etc.
“It should be noted that the first months on dialysis are a period of very high risk, although the role that the start of dialysis plays in itself is not known.” Again is this related to patients who start dialysis in General or those who have DKD as a cause of needing KRT? Would help to clarify if in the paragraph to ensure it is for DKD patients.
“It is important to be able to predict which patients are at the highest risk of mortality and which ones are more likely to progress to KF, as they will be the ones who will really benefit from this multidisciplinary management.”
It can be argued that multidisciplinary care / teams can benefit both groups. I would suggest following on from the initial comment of later referrals and state that they need multidisciplinary care in general. Multidisciplinary care can also involve palliative care teams (should involve) but higher risk for mortality does not preclude them from multidisciplinary care, in fact they may need more of it.
Overall in the introduction:
A lot of shifts in focus here from later referrals to therapeutics for diabetes (which may not be the focus of this review).
Would clarify the focus of this "review" and I would suggest calling it a review rather than “revision”.
Would clearly define the goals of this review so that the rest of the manuscript can be evaluated based on what goals there are. The paragraph describing what the authors hope to review is not clear and frequently shifts from one topic to another.
Heading 2.
Page 2:
Mention of “Trajectories”?
What are those different trajectories? and incidence is low, but DM has been described by the authors as the most common cause of requirement for KRT?
Line 2 of this section KF
KF is defined as by the need for dialysis or CKD? It is unclear here.
Kidney failure and kidney disease is used interchangeably. This needs to be specific.
Review is more focused on type II diabetics than Type I so could consider removing type I information to give that space to in depth review of other topics.
Page 2, Para 5
Very broad statement given in the paragraph and several studies have shown that reducing albuminuria significantly reduces progression of CKD. In fact this is the underlying physiological basis of using Angiotensin Converting Enzyme inhibitors or Angiotensin Receptor blockers and newer therapeutics that are coming out. It may not be a sensitive or specific biomarker however it does corelate with progression of DKD/ CV complications as stated by the Author further down in the article.
I am not sure if the point being made here is whether albuminuria and following that is useful or not if the aim is to slow down CKD. The Author's arguments actually show that it is useful to slow it down.
Page 3 and 4:
While These paragraphs does point to different risk factors for CKD progression it heading is " slow DKD progression and it does not offer strategies or methods on how it can be slowed down in a multidisciplinary way? There is also a mix of discussion on non-modifiable factors and then modifiable factor going back and forth.
There is conflicting evidence regarding glycemic control and worsening of CKD. Once microvascular disease starts whether it is CKD or Retinopathy e.g. tighter glycemic control may not result in reversal or slowing down.
Para 2 Page 4
Word “Molecules”, suggest replacement with pharmacological interventions etc.
Page 4 2.5 Hypertension,
Would suggest referencing the BP cut offs mentioned.
Suggest more info on latest evidence on Dietary Sodium intake in either HTN (2.5) or Diet (2.7).
Sodium intake is the single most important factor in CKD progression and HTN control. Need to emphasize this.
Page 6 Section:
“Prediction of cardiovascular events and progression to KF (G5) in patients with DKD and severely reduced GFR (G4)”
It is not stated as one of the aims of this review. May be when editing the aims paragraph in introduction, this can be added so that it does not come as a surprise to readers if authors do intend to review this very topic.
Before going into “Patient decision making” I suggest clarifying if the aim is to discuss few of them the risk factors in Table 1, although other mentioned in Table 1 are also important.
Section 4 Patient Decision Making:
Line 3: I would suggest mentioning it as “Nephrologists make decision with patients and their families”. It’s a joint decision rather than nephrologists’ decision alone.
Mention of Transition at End of Paragraph: given the focus on being diabetic DKD here the question is: Is that unique to Diabetic Patients only?
Patient activation: “involvement” may be a better word
Page 6 Last Paragraph:
“It is usually not a question of if someone wants a transplant, it is usually more so on dialysis modality and the rest of the questions which are good and important questions”
Page 7 Paragraph 1:
Ending statements about vascular access: “What is unique to patients with DKD in this aspect? These are usual CKD-ESKD care principles in the general population. Authors can talk about success rates of VA in diabetic vs nondiabetics, how many diabetics start with catheter vs VA? Etc.
I would suggest removing the Kidney Pancreas (KP) Transplant discussion. There are criteria for KP Transplant, not everyone qualifies, and modern insulin pumps can also have significant improved outcomes.
KP is always from a deceased donor.
If Authors would like to keep it, then would suggest using the term Deceased Donor Kidney Transplant rather than Cadaveric.
Use of the Word Derivation in Para 6 Page 7? Unclear
Page 8
Would replace "Anyway" with more formal written term than a spoken term
Page 8
Again it is discussing PD options whereas PD should be discussed in the light of Diabetic KD, I am not sure if indications or Contraindications for PD are the aim of this review, it is only adding length to the manuscript that could be better utilized explain why some specific indications/ Contraindications are there to Diabetic Kidney Disease and PD?
Page 9 Para 3:
Unsure what is meant by the “Behavior of patients with Diabetes”?
Page 10 Section 7
Para 3,
All the relevant issues described here, are they more common in DKD than non DKD? Would be helpful to expand on this consistent with the aim of the study.
Page 11 Para 2:
Comparison Paragraph such this should be there on almost all the points discussed under this review. This will highlight how DKD is different than Non DKD which seems to be one of the aims of this review. A Great Paragraph in view of the study aims
I suggest removing the section 9. KRT of the future (bioengineered kidney or implantable bio-artificial kidneys) as I am not sure how that is specific to DKD?
Page 12 Section:
Would suggest referencing the first 2-3 sentences of para 1
Page 13 first paragraph
Would avoid case report references in a review, when there are a lot of studies available comparing outcomes of conservative kidney care and KRT in such populations.
Conclusions:
Unsure what is implied by "but with less evidence", and overall needs some grammatical corrections.
Reviewer 2 Report
I had the opportunity to read the manuscript entitled “Kidney Failure in Patients with Diabetes. What Is the Best Option?” Juan M. Buades et al, who intent to publish in International Journal of Clinical Medicine
From a point of informative medicine, the manuscript is well redacted and focused, but it does not introduce any noticeable new concept for the readers. It appears as a book chapter of medicine or nephrology.
Some minor concerns
This definition is too simplistic: Patients with diabetes mellitus DM) and CKD are called as having diabetic kidney disease (DKD). It must be detailed or expanded. Young readers may consider thst everything is diabetic nephropathy in patients with CKD in which 1mg of glucose is over the upper limit of the normality.
Rewrite: Patients with DKD have a variety of clinical trajectories with different courses and CKD progression to KF. The incidence rate of KF is low. In type 2 diabetes mellitus (T2DM) is 0.29% at 10 years and 0.74% at 20 years from diagnosis of diabetes [8]. In type 1 diabetes mellitus (T1DM) is 2.5 per 1000 person-years [9]. Nevertheless, between 40% and 53% of diabetic people have less severe kidney disease [10], [11].
Are those studies conveniently cited in the text?: SUSTAIN 6, LEADER and REWIND
This paragraph should be further expanded as it introduces new concepts in the disease: Salt has been linked to reduce BP and albuminuria. Dietary sodium considerably modulated the nephroprotective response to RAAS-blockade. While this effect was inde-pendent of blood pressure or antihypertensive co-medication, it was related to pro-teinuria, which persisted in subjects with high dietary sodium intake [37].
Explain better: moderate certainty; along with a reduction in the risk of progression to KF (relative risk, 0.32; 95% CI, 0.18 to 0.56); low certainty.